# Role of Post-Exposure Time in Co(II) Sorption of Higher Concentrations on Electron Irradiated Sheep Wool

**DOI:** 10.3390/molecules24142639

**Published:** 2019-07-20

**Authors:** Jana Braniša, Klaudia Jomová, Renáta Kovalčíková, Peter Hybler, Mária Porubská

**Affiliations:** 1Department of Chemistry, Faculty of Natural Sciences, Constantine the Philosopher University in Nitra, Tr. A. Hlinku 1, 949 74 Nitra, Slovakia; 2Research & Development Laboratory, PROGRESA FINAL SK, s.r.o., Ferienčíkova 18, 811 08 Bratislava - Staré Mesto, Slovakia

**Keywords:** sheep wool, electron beam irradiation, cobalt sorption, post-exposure time, Co-complex

## Abstract

Sorption of Co(II) was investigated on natural as well as accelerated electron beam modified sheep wool involving low and high concentrations up to 200 mmol·dm^−3^. The sorption experiments confirmed the dependence of the sorption capacity not only on sorbate concentration and absorbed dose of energy, but also on post-exposure time. Post-exposure heating to accelerate transformation of the wool structure was of no effect on the sorption comparing with a simple storage for a period of 100 days. Under all tested conditions, the sorption maximum was measured for Co(II) concentration of 125 mmol·dm^−3^ and that was assigned to form a Co(II) complex with keratin. This assumption was tested on visible spectra of mixed solutions of Arginine and Co(II) to be a simplified model of Co(II) interaction with keratin. The sorption decrease is associated with generation of cross links between macro-chains through ligands of the Co-complex. The nodal points are a hindrance to diffusion of next ions into the fibers. Also, pH variations of aqueous extracts from the wool samples depending on absorbed dose and post-exposure time indicate complexity of the structural transformation being specific for each dose applied.

## 1. Introduction

Adsorption technique is largely used within various technological processes, very often to remove undesirable contaminants in the environment. However, pre-concentration using adsorption is topical, especially at recovery selected components from exhausted technological solutions. Conventional adsorbents as activated carbon, clays, zeolites and others have been step by step completed by adsorbents of synthetic or natural origin. There is an observable tendency in research to utilize various waste materials for making low-priced adsorbents from them. That is why recent investigation of inorganic adsorbents is published less frequently than synthetic ones or biosorbents. In principle almost all authors have examined optimal sorption parameters as the effect of concentration, pH, adsorbent dose, contact time, temperature and an adequate model of the isotherms. The papers regarding adsorption of cobalt can be found in wastewater from nuclear stations, metal cutting or chemical industry (dyestuff-making, catalyst systems, dehumidifier), etc.

Inorganic adsorbents for removal of cobalt from a waste involve hydroxyapatites [1,2] indicating chemisorption or physisorption as main mechanism. Çiçek et al. [3] removed non-radioactive Co-ions as well as radioactive ^60^Co applying pumice or zeolite 4A. To remove cobalt(II) from aqueous phase, Manohar et al. [4] examined Al-pillared bentonite clay. As far as synthetic adsorbents, for adsorption of Co(II) Gómez et al. used activated carbons prepared by pyrolysis sucrose and the kinetics data indicated multilayer adsorption [5]. Acrylamide or maleic acid radiation grafted on chitosan showed high sorption capacity towards ^60^Co indicating chemisorption [6]. Vinylbenzyl iminodiacetate as the functional ligand for selective adsorption of Co ions over excess of Fe ions is described by Bhaskarapillai et al. [7]. In the field of biosorbents Vilvanathan and Shanthakumar presented native and biochar forms of *Tectona grandis* leaves with a high adsorption potential to remove Ni (II) and Co (II) ions from aqueous solution [8]. Nazari et al. examined interesting biosorption of a ternary system containing Cu(II), Ni(II) and Co(II) using material known as air-filled emulsion [9]. This stable colloidal suspension consists of bovine serum albumin coated bubbles generated by ultrasound. Having a high surface area, the metal uptake by the protein was high and affinity for Cu(II) was higher than that for Ni(II) and Co(II). It was also showed how lemon peel waste can be used for the removal of cobalt ions from aqueous solutions [10]. Kandah et al. reported even on sheep manure as biosorbent for Co(II) and eventual improvement of the sorption capacity achieved treating the manure surface with citric acid [11]. 

In general, mention of biosorbents with animal origin applied for metal ions removal or their preconcentration is rare and nearly limited to sheep wool [12,13,14]. Only two recent papers [15,16] have provided information on sheep wool modified by electron beam as promising adsorbent for some metal cations. When a metal recovery from scrap is concerned, the corresponding concentration range of the metal can be higher than presented in papers dealing with its adsorption. Examining Cu(II) sorption on both wool native and irradiated for high Cu concentrations above 10 mmol·dm^−3^, Porubská et al. [17] showed an atypical run of the sorption isotherms involving maxima and minima. For those higher Cu-concentrations the sorption mechanism was drafted. It was concluded that formation of Cu-complexes with keratin functional groups occurs already on the fiber surface creating clusters. Those are responsible for restriction of the sorbate diffusion into the fiber bulk while this fact depends on Cu-concentration as well as absorbed dose during the irradiation. We suppose that previously such phenomena were not observed for any cations since adsorption experiments were carried out using low cation concentrations below 10 mmol·dm^−3^. As to concentration, comparison of individual sorption results is rather problematic since authors give bath concentrations variously, in molar or mass concentration. In addition, data on sorption/removal efficiency presented in percentage are misleading without the giving of absolute figures. With reference to Cu(II), it is necessary to take into consideration a special character of copper and its tendency to create complexes due to the 3d-electron structure.

That is why, this work is focused on verification of fluctuation in sorption isotherms also for Co(II) cations having similar disposition to generate complexes with keratin groups.

## 2. Results and Discussion

### 2.1. Effect of Lapse from Wool Exposure on Co(II)-Sorption

Electron beam irradiation of sheep wool splits disulphide bonds of keratin with following oxidation of radicals generated and consecutive transformation of the transitive species for up to cysteic acid [18,19]. In this way, in the exposed wool, additional acidic groups are added capable of forming Co-salts and subsequently complexes with available ligands. As found, primary but also secondary structure of the radiation-modified wool varies with time [19]. It is a gradual complex transformation of conformations in the secondary structure initiated by the absorbed dose of energy and linked with chemical changes. Since both factors play an important role in Co(II) sorption, also sorption potential of the wool should be time-dependent following the dynamics of these changes. To confirm this assumption, we performed sorption experiments at various time intervals of 2–65–100 days from the irradiation with the initial Co(II) concentrations in the (50–200) mmol·dm^−3^ range (Figure 1, Figure 2 and Figure 3).

As can be seen from Figure 1, Figure 2 and Figure 3, the related sorption isotherms and their mutual positions were indeed varied with time. The most variable sorption was measured for 2 days after the wool exposure. These data are consistent with content of the S-oxidized groups observed in irradiated wool at a 4-day interval from the exposure [19]. The greatest change in the position of the sorption isotherms can be seen for doses of 410 and 40 kGy (Figure 1, Figure 2 and Figure 3). While, after two days after the exposure the 410 kGy isotherm showed the highest sorption (Figure 1), this gradually decreased over time (Figure 2) and, after 100 days, its corresponding sorption was the lowest (Figure 3). A considerable jump is also observed for the 40 kGy isotherm; from the lowest position after 2 days, it reached maximum after 100 days. The overall trend in the development of sorption by the irradiated samples suggests an initial lower sorption capacity than the non-irradiated wool excepting 410 kGy sample. The sorption of the exposed samples gradually increased and, at the end of the observation time for concentrations over 125 mmol·dm^−3^, exceeded the sorption of the non-irradiated sample (Figure 3). After 100 days, a maximum for concentration of 125 mmol·dm^−3^ is visible on all isotherms (Figure 1, Figure 2 and Figure 3), the least pronounced for 410 kGy sample. We believe that at this Co(II) concentration, extremely favorable conditions have been developed for keratin-Co(II) interaction. Since Co(II) as a weak Lewis acid tends to form complex forms, the amine and hydroxyl groups of keratin can act as the ligands for Co-carboxylate or Co-cysteinate complexes. Whereas the extreme occurs at all absorbed doses, including 0 kGy, it should not be a result of the absorbed dose but the concentration of Co(II), as the amount of the used wool sorbent was constant. To verify our hypothesis, we used Arginine to be a simplified model of complexing partner for Co (II) cation, since Arginine (Figure 4) is abundantly present in wool keratin in amount around 10% [20].

Arginine contains acidic group -COOH where Co(II) as the central ion replaces H^+^, but also basic ending guanine-group (-NH-C(=NH)-NH_2_), capable to provide non-binding electrons for coordination. Thus, Arginine with Co(II) may create an intrinsically complex salt, a chelatonate.

Arginine and Co(II) in mixed solution with 1:1 or 2:1 molar ratio were allowed to react, and time development of the reaction was monitored by VIS spectrometry. The time-dependent VIS spectrum is displayed in Figure 5.

While the CoCl_2_ spectrum itself is simple and smooth with λ_max_ = 512 nm (not displayed), the spectra of the mixed solutions are mildly rugged up to the interval of 6 h from the mixture (Figure 5) indicating generation of transitive structures. Excepting the first 15 min, the absorbance of the 2:1 sample around 500 nm is always higher compared to the solution 1:1, although the Co(II) portion in the sample 2:1 is lower and, alone prevailing Arginine does not absorb in visible region at all. The shape variations of the spectra and the λ_max_ shifting characterize progressive building of a complex. Here, compared to the CoCl_2_ spectrum, soft batochromic and hypsochromic but, namely hyperchromic shifts are observable and, the solution 2:1 of Arginine prevalence shows larger variations.

In principle the model reaction of Co(II) with Arginine can be a certain simplified simulation for the interaction of Co(II) with acid- and amine-groups in wool keratin when are positioned favorably. However, if in (non-)irradiated wool owing to any reasons, mainly spatial or structural ones, some ligand coming from other keratin chain occurs within force field of Co(II) carboxylate or cysteinate, the ligand can become constituent of the complex. In such case such crosslinking involves two or even more macro-chains. Similar situation arises when Co(II) reacts with two acid groups bound to different keratin chains by substitution mechanism. Chemical nodes shaped in such way obstruct diffusion of next sorbate into the fibers. Based on FTIR spectra the same fact has been documented for Cu(II) sorption on wool, too [17]. In addition, as mentioned by Zhang et al. [21], depending on the different coordination ability of the amino acids towards Co(II) based on different pH of medium, formation of the complexes can be various. As regards to adsorption studies, those reasons have been mentioned only rarely, if any. The aim is mostly to find optimal conditions enhancing sorption capacity without a deeper reasoning.

The decrease in sorption beyond the maximum for 125 mmol·dm^−3^ and a further slight rise/fall demonstrates the dynamics of transformation of the intermediates corresponding to the individual absorbed doses. However, the main reason for the observed course (Figure 3) is the formation of complex/complexes at higher Co(II) concentrations, which is not possible at insufficient concentration. The formation of complexes already on the fiber surfaces means hindered access of other Co(II)-ions inside the keratin fibers due to the mentioned cross-linking of the macro-chains. The Co(II)-ions can overcome such barrier for diffusion only due to increasing concentration. In comparison with Co(II), in the study of Cu(II) sorption, which is highly prone to form complexes, several sorption extremes on the irradiated wool were observed within the concentration of (12–80) mmol·dm^−3^ [17]. This points to the different coordination properties of Cu(II) and Co(II). As reported in papers [22,23,24], the spatial arrangement of Cu-complexes is variable according to environmental conditions. Therefore, several maxima on the sorption dependence may correspond to several types of polyhedron. Similar variability for Co(II) was not observed in our conditions.

If the formation of the Co(II)-Arginine complex is time-dependent (Figure 5), even more complicated course of the interaction must be in the wool. The changing process of Co(II)-sorption illustrates the relative time developing of the sorption compared to the sorption 2 days after the exposure (Figure 6).

While sorption on the non-irradiated wool (0 kGy) did not change over time, the exposed wool showed time variations depending on both dose and Co(II) concentration. The sorption for samples with doses of (21–153) kGy achieved a higher sorption after 100 days than after 2 days, respectively, and the same could be said – except for 21 kGy sample – on 65-day´s lapse, too. Another situation appears for the samples with doses of 258 and 410 kGy; the corresponding sorption is smaller than after 2 days, with a lower level for 410 kGy sample. Based on our above mentioned observations, the wool with lapse of 100 days after the exposure and with absorbed dose of (40–153) kGy appears to be suitable for Co(II) sorption from bath with concentration above 50 mmol·dm^−3^.

Most authors, if not all, examined (ad)sorption of Co(II) only for low (ad)sorbate concentrations. We also tested the sorption capacity of the irradiated wool for low Co(II) concentrations in the range (0.2–0.8) mmol·dm^−3^ after 100 days (Figure 7).

Of course, lower order Co (II) bath concentrations showed a correspondingly lower sorption. The isotherms for both the non-irradiated wool and the wool with different absorbed doses are virtually identical for such low concentrations. Our explanation is that a small amount of Co(II) is not enough to form Co(II) complexes and, the original carboxyl groups of keratin are sufficient even without the participation of cysteine forms in the irradiated samples. Therefore, no significant differences in the sorption are observed. Similar results for sorption of Pb(II), Cr(III) and Cd(II) on both irradiated and non-irradiated wools for concentrations up to 0.4 mmol·dm^−3^ were also obtained by Hanzliková et al. [16].

### 2.2. Effect of Post-Exposure Wool Heating on Co(II)-Sorption

Post-exposure heating of the wool prior to sorption was motivated by an assumption that the heating could accelerate transformation of the intermediates to final cysteine acid. The wool with the least stable structure, i.e., of 2-day´s lapse from the irradiation, was heated to 40 °C or 60 °C in laboratory oven for 4 h, 8 h and 24 h. Then, these samples were subjected to the sorption experiments with the initial bath concentration of 150 mmol·dm^−3^. The sorption results were compared with the sorption corresponding to the unheated sample (Figure 8).

Figure 8 shows that the heating effect depends on the absorbed dose. Heating at 60 °C for 4 h was more effective for doses up to 153 kGy, and the heating at 40 °C was more effective for higher doses. The practical importance of the heating appears only for doses of 40 and 153 kGy with almost the same effect for both temperatures. The heating at 40 °C for 8 h improved sorption only for the wool with doses of 40 and 153 kGy. The other samples showed impaired sorption for 40 °C. All samples had worse sorption when heated to 60 °C, which is attributed to a gradual wool denaturation. The heating at 40 °C for 24 h showed a positive effect only on 153 kGy wool and the sorption improvement was practically the same as the heating at 40 °C for 4 h. Milder improvement in the sorption after the heating to 60 °C were observed only for 21, 40 and 153 kGy.

Period of 100 days was preliminary assessed as sufficient to transform the intermediates into a final form. To obtain information to what measure the post-exposure heating wool with 2-day´s lapse is equivalent to 100-day´s lapse, we compared Co(II) sorption on the unheated wool with 100-day´s lapse with the sorption on the wool heated 2 days after the exposure (Figure 9).

The results of the comparison showed that the sorption of none from the heated exposed samples reached the sorption level of the unheated wool with 100-day´s lapse. The only, but only slight improvement was achieved by heating the native wool at 60 °C for 4 h. Only 410 kGy sample showed an equalizing the effect of the 8 h heating at 40 °C with 100-day´s lapse from the exposure. All other heat treatments worsened the sorption. The 24 h-heating at 60 °C resulted in a slight sorption improvement only for 21 kGy sample. None of the heat treatments achieved the effect of 100-day´s simple storage of the samples. Thus, the heating of the wool did not accelerate the transformation processes as expected.

Corresponding Co(II) sorption is an indirect summary indicator of processes passed involving, besides the chemical transformation of S-oxidized groups, changes in the configuration of keratin chains. The finding that conditioning wool at higher temperatures did not bring any significant improvement in sorption capacity is beneficial for potential practical applications, as simple storage without handling will simplify the manipulation and will not require energy consumption.

### 2.3. Effect of Lapse from Wool Exposure on pH of Aqueous Extract

The pH of medium has a considerable effect on the Co(II) sorption but, not only of this cation [25]. In general, the pH value of protein environment is important factor because it determines value of isoelectric point varying around the pH~4 [26]. To sorb any cation effectively, the process has to be conducted at pH above the wool isoelectric point. In such case the positively charged cation is not repelled by the positively charged surface but readily interacts with the negatively charged fiber surface. In order to better understand the Co(II) sorption process on the irradiated wool, we measured pH value of the aqueous extract of the wool samples after 24 h wool contact with deionized water at time intervals consistent with the measured Co(II) sorption. In our case, the pH of deionized water with initial value of 6.28 was increasing after contact with all wool samples and at each time interval from the exposure, although not monotonously (Figure 10).

The largest changes in the pH of the aqueous extract showed wool samples at 65 days from the exposure, with one peak for 21 kGy and total change of 0.54 pH units. One extreme was also observed for 100 day samples, but for 40 kGy, with total span of 0.46 pH units. The samples with 2-day´s lapse showed qualitatively different pH course; although the overall pH change was the smallest, only 0.28 units, the curve had two extremes, for 21 and 99 kGy. Two peaks for the same doses are also shown by the corresponding sorption after 2 days and both dependencies go quasi in parallel up to 99 kGy and, reversely from 153 kGy. The 65-day´s lapse with one peak showed qualitatively mirror image for both dependencies up to 258 kGy dose. At the 100-day´s lapse is observed a small mutual shift in the maximum for both pH and sorption parameters (see 21 and 40 kGy), but only in this case, a reverse course did not occur; the both dependencies indicate a quasi-parallel development. Thus, with increasing lapse, the differences in character of the pH aqueous extracts and the Co(II) sorption are reduced. For the 100-day´s lapse it can be already said that the lower the pH, the lower the sorption. This fact is consistent with accepted view that the higher pH of isoelectric point, the more negative surface charge of the wool and the higher attraction for cations. Also, Wen et al. [25] presented higher Co(II) adsorption on wool at pH~10 than at pH~8. However, when increasing the pH, the limiting factor for the sorption is possible precipitation of related cation from the solution.

The increasing pH in low dose wool extracts compared to the pH of deionized water means increasing concentration of OH^-^ and decreasing H^+^, and vice versa. We attribute the decrease of H^+^ in the extract to its binding by negatively charged groups of wool such as cystine oxides (cystine monoxide -SO-S- and cystine dioxide –SO_2_-S-) and S-sulfonate (R-S-SO_3_^−^), which corresponds to low absorbed doses being precursors of cysteine acid R- SO_3_H [18,19]. As reported by Oae and Doi [27], cystine oxides are strong hydrogen acceptors, reducing the presence of free H^+^ in solution. In this case, the amount of OH^-^ must be predominant in the solution to keep the ionic product of water. Vice versa, acidification of the media is due to release of H^+^ as a result of dissociation of the acidic groups initially present (R-COOH) as well as resulting from running transformation processes (R-SO_3_H). Summarized, the development of the pH of the extract (Figure 10) is a function of the corresponding absorbed dose and the wool chemical structure at a given time.

## 3. Materials and Methods

### 3.1. Materials

Analytical grade reagents were provided from Centralchem (Bratislava, Slovakia).

Stock solution of Co(II) cation was prepared by dissolving appropriate amount of cobalt salt CoCl_2_·6H_2_O in deionized water. Testing solutions for sorption experiments were prepared by diluting the stock solution in deionized water to desired initial concentrations.

The standard solution of Co(II) for AAS calibration (1000 mg·dm^−3^ in 3% HNO_3_) were supplied by Agilent Technologies (Santa Clara, CA, USA). Nitric acid 67% ANALPURE^TM^ for trace analysis to dilute the standard and wash the AAS spectrometer was obtained from Analytika (Prague, Czech Republic).

l-Arginine 98% was supplied by Alfa Aesar (Karlsruhe, Germany).

The raw sheep wool came from spring sheep-shearing (2018) of a Merino-Suffolk crossbreed bred in West Slovakia. The impurities (dirt, food, dung) adhered to the wool were washed repeatedly with running water until it remained visually clear. The dried wool was then thoroughly cleaned by hand to remove animal and plant residual contaminants adhered to the surface of fiber. The scouring procedure of sheep wool was finished in ultrasonic bath Kraintek K5LE (Podhájska, Slovakia) with tap water tempered to 40 °C for 10 min using 3 repetitions with water exchange. Then the wool was washed with 5 L of deionized water and free-dried at the room temperature and humidity for 3 days. The wool was stored in a paper box for following electron beam irradiation.

### 3.2. Electron Beam Irradiation

The irradiation of the wool stored in paper boxes was performed in linear electron accelerator UELR-5-1S of Progresa Final SK operator (manufacturer FGUP “NIIEFA“, Petersburg, Russia). The conditions of the exposure are presented in [18]. The doses applied were 0–21–40–99–153–258–410 kGy. After being irradiated the samples were kept under usual laboratory conditions.

### 3.3. Sorption Experiments

The sorption experiments were conducted with Co(II) solutions applying concentrations in the range of (0.2–0.8) and (50–200) mmol·dm^−3^, respectively, what corresponds approximately to (0.0012 –0.0047 and 0.3–1.18) % of Co. The wool fibres cut to 3–5 mm in amount of 0.2 g were immersed in 20 cm^3^ of the testing solution in a small glass cup with a cap and shaken for first 6 h at room temperature on a laboratory horizontal shaker Witeg SHR-2D (Labortechnik, Germany) at 100 rpm, and then kept in static mode for next 18 h. Then the contact solution was filtered through KA5 filter paper and used for determination of residual concentration of Co(II). Every sorption experiment was carried out in triplet. 

The sorption capacity was calculated using the following equation:$$S=\frac{x_{1}-x_{2}}{m}$$
where *S* is the sorption capacity defined as the mass of sorbate in mg per 1 g of the sorbent for individual wool samples when particular testing solution is applied in specified concentration,
*x*_1_ is the mass of Co(II) added in the initial solution (mg),*x*_2_ is the residual mass of Co(II) in the solution after its contact with the wool sample (mg),*m* is the mass of wool sample taken for analysis (g).

### 3.4. Spectral Measurements

Atomic absorption spectrometry (Spectrometer 240 FS AAS, Agilent Technologies, Santa Clara, CA, USA) was utilized for sorption experiments applying low concentrations of Co(II) in the range 0.2–0.8 mmol·dm^−3^. The operative parameters were as follows: wavelength emitted by lamp of 324.8 nm, flow acetylene/air of 2.0/13.5 dm^−3^·min^−1^, concentration of Co(II)-calibration solutions of 5–10–15 mg·dm^−3^, solution used to optimize absorbance Co-signal to 0.2 with concentration of 2.5 mg·dm^−3^.

The Co(II) of higher concentration applied in the sorption experiments (Section 3.3) was determined using visible spectrometry Specord 50 Plus (Analytikjena, Germany) with 1 cm cell. Concerning spectra, deionized water extract from corresponding wool sample obtained under identical conditions such as Co(II) residual solution was used as blank for each series of the samples with identical absorbed dose. The taken spectra were evaluated by means of corresponding calibration curves. The Co(II) spectrum showed maximum at λ = 512 nm. VIS spectrometry was used also for observation of model complex Arginine-Co(II) development.

### 3.5. Measurement of pH 

Value of pH in the solutions was measured using pH-meter Orion2 Star (Thermo Scientific, Waltham, MA, USA) equipped with plastic electrode Sen Tix 42 with temperature sensor. Applying double measuring the relative error did not exceed ± 0.5%.

## 4. Conclusions

The work studied the character of Co(II) sorption on wool natural and modified by accelerated electron beam. Test concentrations included low (from 0.2 to 0.8 mmol·dm^−3^) as well as high concentrations (from 50 to 200 mmol·dm^−3^) of Co(II). The sorption experiments performed for those higher concentrations at different time intervals from the wool irradiation confirmed that also the time changes of the wool structure have an impact on the sorption capacity of this sorbent. Level of the sorption changed not only with the sorbate concentration, but also with the absorbed dose and with time. The highest sorption increase, of almost 550% compared to the 2-day´s lapse from the exposure, was recorded for 153 kGy for Co(II) concentrations of 200 mmol·dm^−3^ after 100 days. Conversely, under these conditions, below the comparative data for 2 days, the sorption for the 410 kGy sample fell to 55%. Post-exposure the wool heating to 40 or 60 °C for 4–24 h in order to accelerate the transformation processes did not bring any advantage in increasing sorption levels compared to the 100-day´s wool storage under normal conditions.

Under all conditions tested the sorption dependencies for higher concentrations showed a maximum corresponding to Co(II) concentration of 125 mmol·dm^−3^, which was attributed to the formation of a Co(II) complex. As suggested by the time changes in the VIS spectra of mixed solutions of Arginine and Co(II) as a simplified model for interaction of Co(II) with keratin, the formation of such complex is highly realistic in the wool, too. While in spite of favourable conditions in solution the formation of the complex is gradual here, in the wool it can proceed slowly, not only due to the hindered steric conditions, but also the simultaneous transformation of S-oxidized structures into the final cysteine acid and the conformational changes. The formation of the complex already on the surface of the fibres leads to cross-linking of the keratin chains through the ligands. Such nodes make difficult for other Co(II)-ions to enter the fibre bulk, which reduces the sorption. 

Changes in the pH of aqueous extracts prepared from the wool samples at the lapses from the exposure confirmed relation with the structural changes in the wool induced by the irradiation. The increasing time lapse indicated a positive impact on the stabilization of trend of these variations.

## Figures and Tables

**Figure 1 molecules-24-02639-f001:**
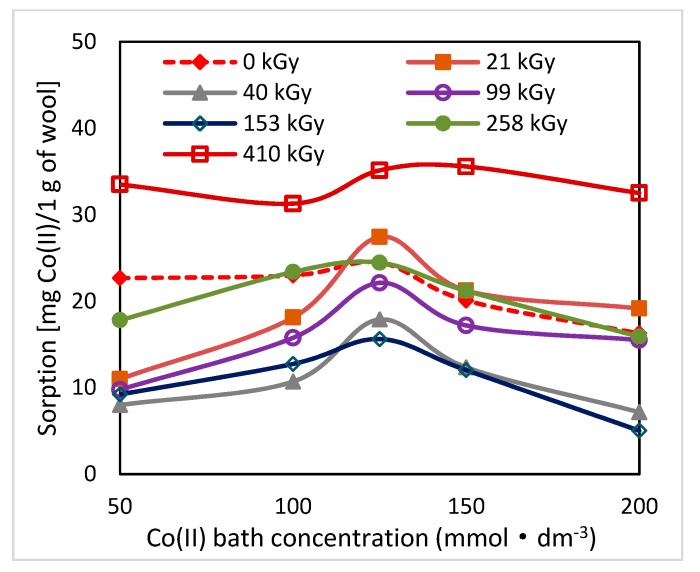
Variation of sorption of Co(II) on the wool samples depending on the initial bath concentration measured 2 days after the irradiation.

**Figure 2 molecules-24-02639-f002:**
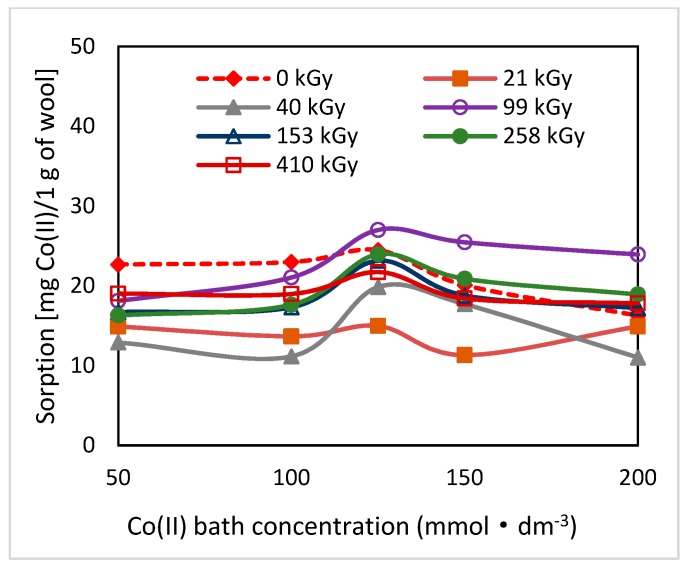
Variation of sorption of Co(II) on the wool samples depending on the initial bath concentration measured 65 days after the irradiation.

**Figure 3 molecules-24-02639-f003:**
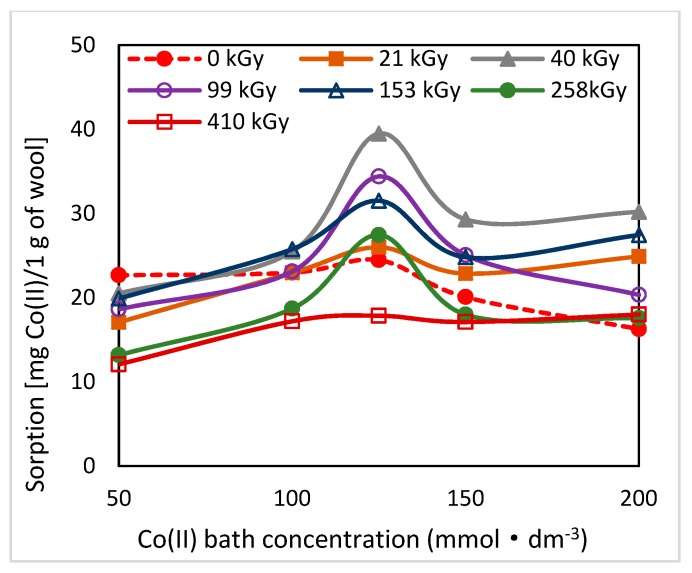
Variation of sorption of Co(II) on the wool samples depending on the initial bath concentration measured 100 days after the irradiation.

**Figure 4 molecules-24-02639-f004:**
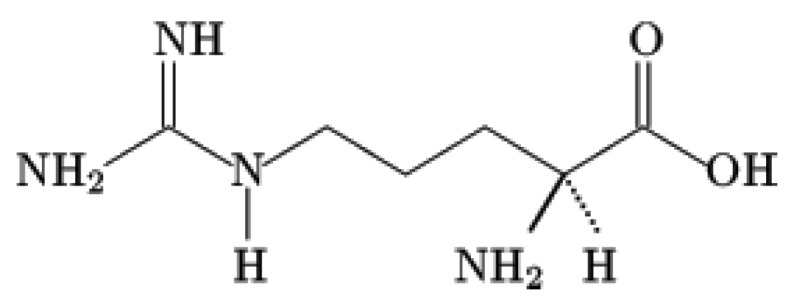
Structure of Arginine.

**Figure 5 molecules-24-02639-f005:**
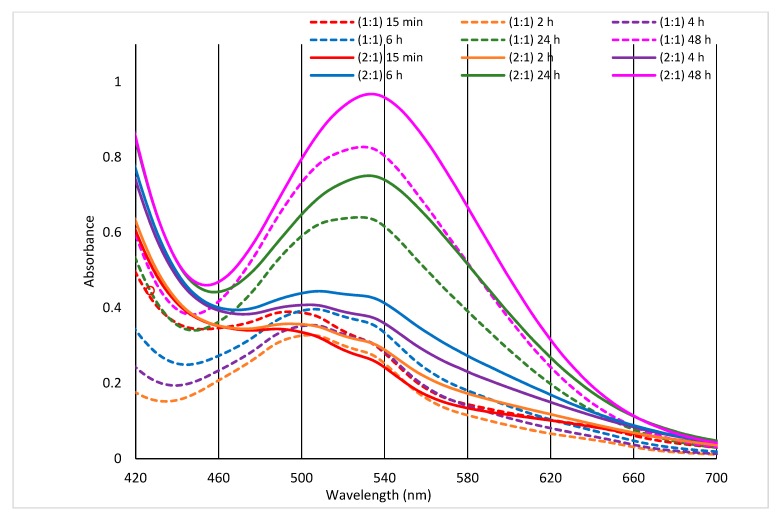
Timing structure development of Arginine-Co(II) complex starting from mixing the components; solid curves represent mixture solution in molar ratio Arginine to Co(II) = 2:1 and dashed ones 1:1, respectively.

**Figure 6 molecules-24-02639-f006:**
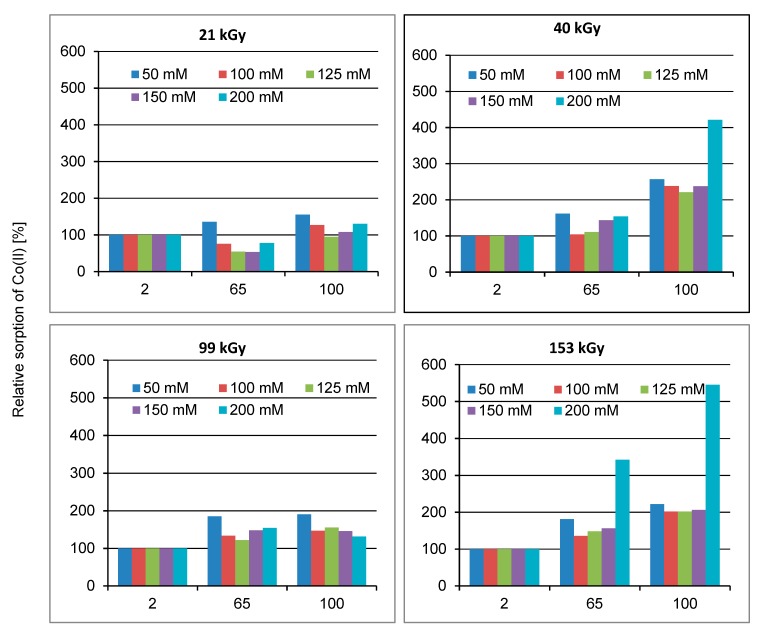
Relative time variation of Co(II) sorption on the wool compared with sorption for 2-day’s lapse from the exposure.

**Figure 7 molecules-24-02639-f007:**
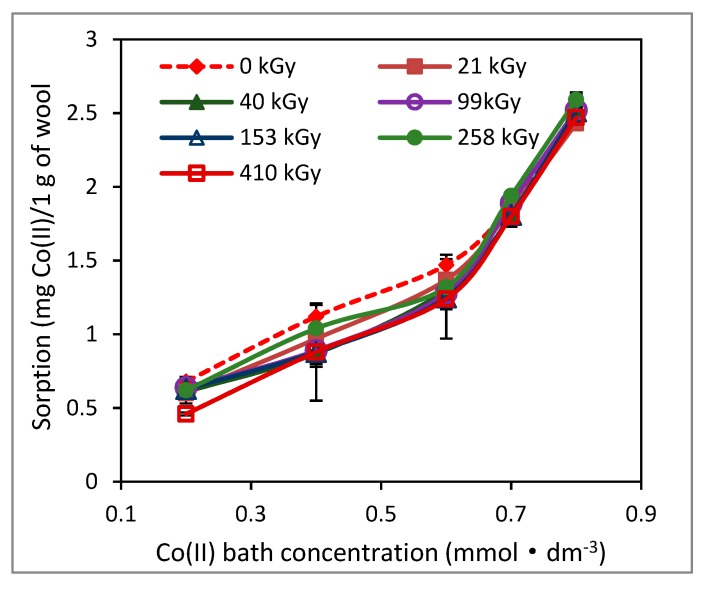
Variation of Co(II) sorption on wool with different absorbed dose depending on Co(II) concentration in bath after 102 days from irradiation.

**Figure 8 molecules-24-02639-f008:**
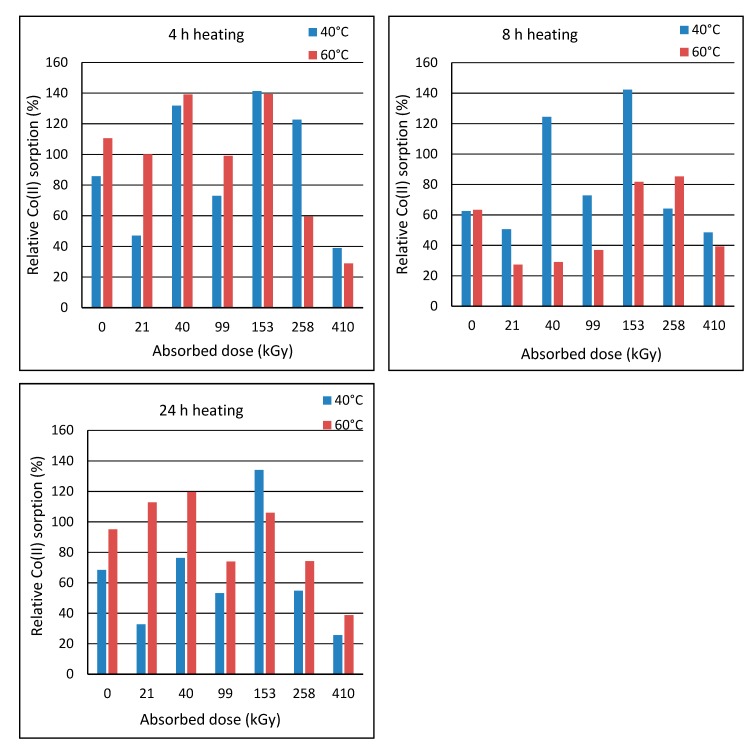
Co(II) sorption on the wool after 2 days from exposure and pre-heated to 40 or 60 °C for 4-8-24 h compared to the unheated wool; initial concentration of 150 mmol·dm^−3^.

**Figure 9 molecules-24-02639-f009:**
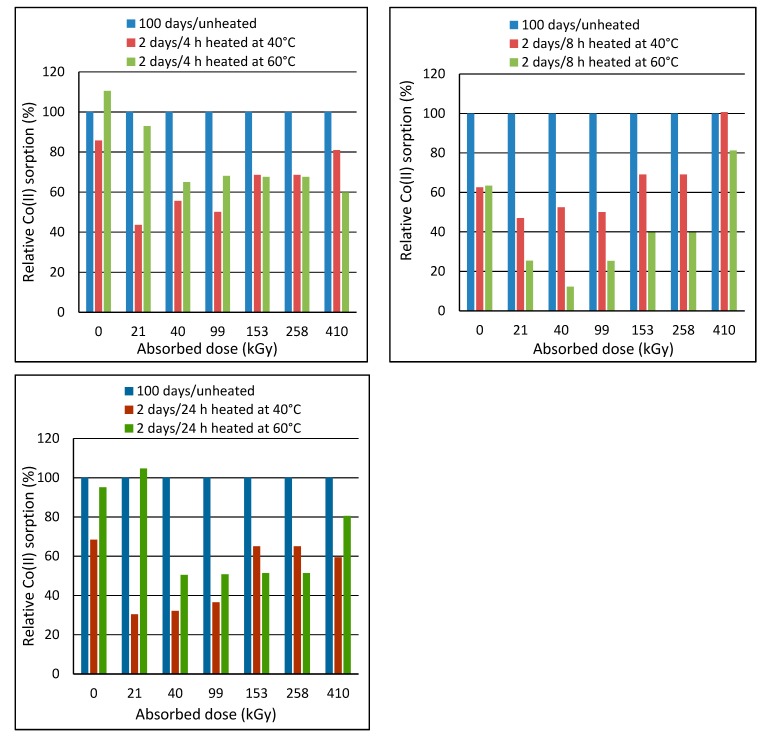
Comparison of Co (II) sorption efficiency on wool with 100-day´s lapse from exposure to wool with 2-day´s lapse and pre-heated at 40 or 60 °C for 4–8–24 h; initial concentration of 150 mmol·dm^−3^.

**Figure 10 molecules-24-02639-f010:**
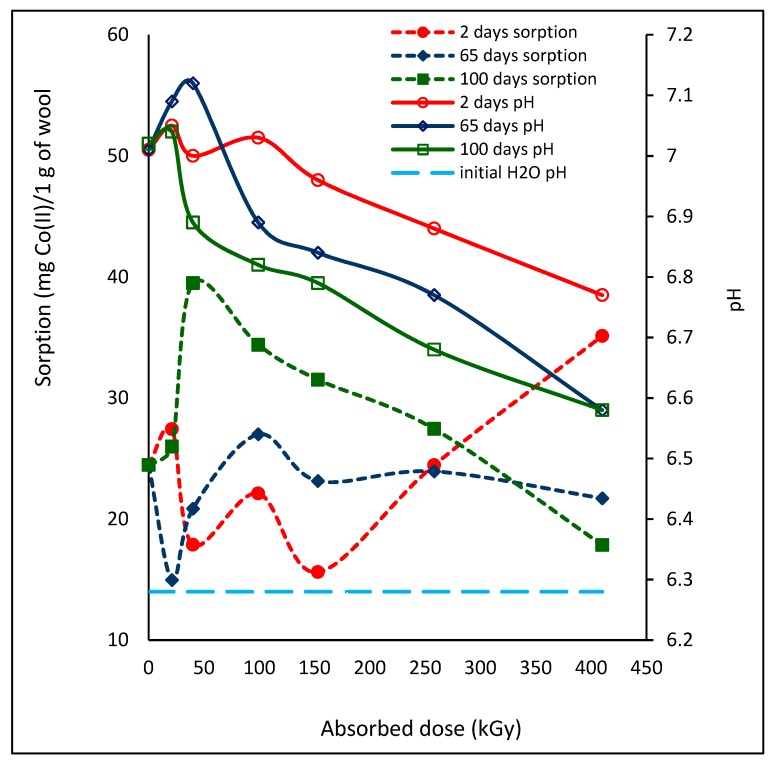
Variation of Co(II) sorption (for c = 125 mmol·dm^−3^) and pH of aqueous extracts from wool with different time lapse from the irradiation depending on absorbed dose.

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
