# Peer review of "Role of Post-Exposure Time in Co(II) Sorption of Higher Concentrations on Electron Irradiated Sheep Wool"

_molecules, 2019, doi:10.3390/molecules24142639_

Round 1
Reviewer 1 Report
This manuscript prepared by J. Braniša and co-workers reports interesting studies concerns sorption of Co(II) of low and high concentrations onto sheep wool natural and irradiated by electron beam. Authors also studied effect of post-exposure heating of wool on Co(II)-sorption and effect of lapse from wool exposure on pH of aqueous extract. Overall, I think there are clearly enough new and interesting results for the publication in Molecules, but after but after having some small corrections.
1. In Introduction part in 40 line is “apllying” but should be applying.
2. Figures 1-3: please use the same marking of curves on charts. E.g for 0 kGy on Figure 1 and 2 is dashed line with rhomb but on Figure 3 with circles. Please check it and correct to be the same marking on all Figures.
3. Figure 5 is a bit confused, I mean marking. Authors used dashed, dotted and whole lines, and in description of this Figure used additional marking. Please change it, use e.g dashed line with circles, the same as you did on Figure 10.
Author Response
Dear reviewer, thanks for your recommendation. We accepted all. We believe that your comments will help us at future studies. Our responses are as follows:
Reviewer´s comment | Authors´reply |
1. In Introduction part in 40 line is “apllying” but should be applying. | This typing error has been corrected. In addition, other minor adjustments have been made and they are highlighted. Done. |
2. Figures 1-3: please use the same marking of curves on charts. E.g for 0 kGy on Figure 1 and 2 is dashed line with rhomb but on Figure 3 with circles. Please check it and correct to be the same marking on all Figures. | The reviewer is right. The marking has been checked and modified to be consistent. Done. |
3. Figure 5 is a bit confused, I mean marking. Authors used dashed, dotted and whole lines, and in description of this Figure used additional marking. Please change it, use e.g dashed line with circles, the same as you did on Figure 10. | Thank for this comment. The original marking in Figure 5 was rather complicated to reader. The figure has been now simplified. Thanks to colour it is easy reader orientation also without markers. Done. |
Reviewer 2 Report
This paper actually is a continuation of the research work of authors team in the field of irradiation impact on the sheep wool in the presence of transition metal complexes. Particularly in the recent paper (https://doi.org/10.3390/molecules23123180) authors have studied the anomalous sorption of Cu(II) ions by sheep wool at high doses of irradiation. In the current paper the Co(II) sorption is studied in the similar manner. I have not significant remarks to this paper except the quality of figures. I think that the grey grid should be removed from all the spectra and some histograms. It seems that all these figures are just inserted from Excel program, while the professional software like OriginLab should be used for the spectra preparing. Also, the axis captions are incorrect in some places (mmol.dm-3 - the dot below is not a multiplication sign; Wave-length (nm) - should be Wavelength without -). Finally, I recommend the acceptance of the MS after the minor revision in part of figures preparation.
Author Response
Response to reviewer 2
Dear reviewer, thanks for your recommendation. We tried to accept them according to our capabilities. We believe that your comments will help us at future studies. Our responses are as follows:
Reviewer´s comment | Authors´reply |
1. The grey grid should be removed from all the spectra and some histograms. | The grey grids have been removed from all figures. Done |
2. It seems that all these figures are just inserted from Excel program, while the professional software like OriginLab should be used for the spectra preparing. | The reviewer is right; the figures are constructed using Excel since our laboratory does not have the official OriginLab software available. If the excel images are unacceptable, we would have to ask another workplace for cooperation. In that case, we would need at least two more days and, to complete the Acknowledgment in the manuscript. |
3. Also, the axis captions are incorrect in some places (mmol.dm-3 - the dot below is not a multiplication sign; | The axis captions were corrected following the comment. Done |
4. Wave-length (nm) should be Wavelength without -). | The axis caption has been modified to Wavelength. Done |